# Association between E-Cigarette Advertising Exposure and Use of E-Cigarettes among a Cohort of U.S. Youth and Young Adults

**DOI:** 10.3390/ijerph191912640

**Published:** 2022-10-03

**Authors:** Vuong V. Do, Amy L. Nyman, Yoonsang Kim, Sherry L. Emery, Scott R. Weaver, Jidong Huang

**Affiliations:** 1Department of Population Health Sciences, School of Public Health, Georgia State University, Atlanta, GA 30303, USA; 2Tobacco Center of Regulatory Science, School of Public Health, Georgia State University, Atlanta, GA 30303, USA; 3NORC at University of Chicago, Chicago, IL 60637, USA; 4Department of Health Policy & Behavioral Sciences, School of Public Health, Georgia State University, Atlanta, GA 30303, USA

**Keywords:** e-cigarettes, advertising, marketing

## Abstract

Youth and young adult (YYA) use of e-cigarettes increased rapidly between 2010 and 2019 in the United States, during which exposure to e-cigarette advertising was also increased. We aimed to examine whether exposure to e-cigarette advertising among YYAs may lead to subsequent e-cigarette use. A cohort of 3886 YYAs ages 13–24 was recruited from two survey panels in 2018 and followed up until 2019. Survey data were collected online and by telephone. The primary outcome was past 30-day use of e-cigarettes at the follow-up survey. Among 2304 YYAs who retained at the follow-up survey and were not past 30-day e-cigarette users at baseline, both youth and young adults exposed to e-cigarette advertising at baseline had elevated odds of past 30-day e-cigarette use at follow-up (Youth adjusted odds ratio (aOR): 2.77, 95% CI: 1.23, 6.24; Young adults aOR: 2.34, 95% CI: 1.08, 5.11) compared with those not reporting baseline advertising exposure. The majority of YYAs reported exposure to e-cigarette advertising at baseline (Youth: 63.7%, 95% CI: 59.8, 67.4; Young adults: 58.3%, 95% CI: 53.6, 62.8). Our findings suggest that exposure to e-cigarette advertising was associated with an increase in subsequent past 30-day use of e-cigarettes among YYAs. Restricting advertising targeted at YYAs may reduce the likelihood of e-cigarette use among YYAs.

## 1. Introduction

Between 2010 and 2019, the use of e-cigarettes has increased dramatically among youth and young adults (YYAs) in the United States [1,2,3,4,5,6,7], alarming tobacco control researchers and policymakers. In 2018, the U.S. Surgeon General declared the use of e-cigarettes among youth an epidemic [8]. Numerous strategies have been proposed to combat the increase in e-cigarette use among youth, including, but not limited to, restrictions on the sale of flavored e-cigarettes [9], price increases, restrictions on promotions, and implementing inclusive smoke-free policies [7]. In December 2019, the Congress raised the minimum age for purchasing tobacco products from 18 to 21 to reduce and prevent youth e-cigarette use. Prior to that, many U.S. states and territories had already enacted laws increasing the minimum legal tobacco purchase age to 21 [10].

Exposure to e-cigarette marketing and advertising has increased among youth in recent years [11] and prior research suggests such exposure may be associated with perceptions of, interest in, and use of e-cigarettes among youth and young adults [12,13,14,15,16,17,18,19,20,21,22,23,24,25,26,27,28,29,30,31,32,33,34,35,36,37,38,39,40]. The majority of previous studies on e-cigarette advertising utilized either a cross-sectional [12,13,14,15,16,17,18,19,20,33,36,39,40] or experimental design [21,22,23,24,25,26] to assess associations between advertising and e-cigarette perceptions, susceptibility, or use. Among the cross-sectional studies, many utilized National Youth Tobacco Survey (NYTS) data on reported advertising exposure via several media channels and e-cigarette use, or intentions to use [12,14,15,16,17,18,19,39]. Prior experimental studies often manipulated exposure to one specific channel of advertising to establish associations with e-cigarette advertising and intentions to use [22,23,24,25,26]. Although these studies provided foundational evidence on the potential link between exposure to e-cigarette advertising and product use, cross-sectional designs cannot address a temporal relationship, and experimental studies can only assess the intention to use, not the actual behaviors. Thus, longitudinal studies are needed to examine the potential temporal relationship between marketing exposure and subsequent e-cigarette use. To date, several cohort studies have been conducted, most of which assessed the association between exposure to advertising through one specific media channel [27,28,30,31,32,35,37,38], and did not examine the potential differences by media channels or locations of exposure. In addition, many of these cohort studies did not account for potential confounders, such as cigarette use, in their analysis.

Additionally, much of the prior e-cigarette advertising research involves only youth aged 11–17 years [12,13,14,15,16,17,18,19,21,22,23,24,25,28,29,30,32,33,38,39,40], with few examining the impact of e-cigarette advertising specifically on young adults aged 18–29 years [20,26,31,35,37]. Chen-Sankey et al. examined a cohort of both youth and young adults to assess the impact of advertising exposure on e-cigarette use, finding a relationship between marketing exposure and e-cigarette experimentation [27]. However, this study relied on data from 2014 through 2016. Similarly, Loukas et al. studied cohorts of YYA college students and found associations between baseline ad exposure and e-cigarette initiation in 2017 [34]. Given the significant increases in e-cigarette use among YYAs since 2017 [1,4,6], and the proliferation of e-cigarette marketing campaigns in the past few years [41,42,43,44], more updated cohort studies are needed.

Although prior research has documented a difference in e-cigarette advertising exposure by age group [45], few studies have examined whether the association between e-cigarette advertising exposure and e-cigarette use may differ between youth and young adults. Youth may be more susceptible to the influence of advertising and may also be impacted differently by different media channels, creating the need to assess these age groups separately. Our study considers youth and young adults separately, comparing them on demographic characteristics to assess the associations with marketing exposure and the product use.

To address these gaps in prior research, this study utilizes a longitudinal design, examines and compares associations between advertising exposure and e-cigarette use for both youth and young adults, and includes exposure via three important channels of advertising exposure—television, point-of-sale, and online/social media. Specifically, we examine whether exposure to e-cigarette advertising at baseline via these media channels was associated with past 30-day e-cigarette use one year later among a cohort of American youth and young adults. We hypothesize that, other thing constant, youth and young adults who reported exposure to e-cigarette advertisements at baseline would be more likely to become subsequent e-cigarette users compared with those who did not report e-cigarette advertising exposure.

## 2. Materials and Methods

### 2.1. Study Design and Participants

Participants aged 13–24 were recruited from two nationally representative survey panels—NORC’s AmeriSpeak^®^ Panel and GfK’s (now IPSOS) KnowledgePanel. NORC’s AmeriSpeak is a probability-based nationally representative panel, with households selected from a sample frame. Randomly selected households were sampled with a known, non-zero probability of selection from the NORC National Frame and address-based sample, and then contacted by U.S. mail and by NORC telephone and field interviewers. NORC’s National Frame is designed to provide at least 97 percent sample coverage of the U.S. population by supplementing the U.S. Postal Service Delivery Sequence File. To do this, NORC field staff surveyed select geographic areas and created a supplemental list of addresses for the National Frame. This in-person listing of households improves sample coverage from 92 percent (based on address-based sampling) to 97 percent (using the NORC National Frame). In the process, the AmeriSpeak Panel was designed to be representative of the U.S. non-institutionalized population by providing enhanced sample representation of hard-to-reach rural households. KnowledgePanel is one of the largest probability-based online panels in the U.S. It was built on a foundation of address-based sampling (ABS) and provides a statistically valid representation of the U.S. population as well as many under-researched and often harder-to-reach populations. KnowledgePanel’s sample design uses a single sampling frame: the Delivery Sequence File (DSF) of the United States Postal Service, which covers almost 100% of the U.S. population. A random sample of households from across the United States were sent a mail invitation to join the KnowledgePanel. For this study, we used both panels because neither panel alone could provide sufficient YYA sample (ages 13-24) based on our power calculation.

Data collection occurred during April–June, 2018 for the baseline survey. A total of 3886 youth and young adults (ages 13–24) were surveyed at baseline. Cumulative response rates for the baseline survey were as follows: 9.2% for AmeriSpeak adults and 7.0% for AmeriSpeak youth, 3.5% for KnowlegePanel adults and 4.5% for KnowledgePanel youth. A follow-up survey was conducted in June–September 2019, retaining 2555 (65.7%) YYAs from the baseline survey. The analytic sample for this study comprises 1490 youth (ages 13–20) and 814 young adults (ages 21–24) who were retained from baseline and who were not past 30-day e-cigarette users at the time of the baseline survey. The surveys were offered in English on both phone and web for the AmeriSpeak Panel and web only for KnowledgePanel. A written consent form was obtained from all participants. For youth under the age of 18 to participate, a parent or legal guardian needed to provide consent, in addition to youth’ own assent. This study was approved by the Georgia State University Institutional Review Board (approval Number: 344250).

### 2.2. Outcome Variable

The study’s outcome variable was past 30-day e-cigarette use at follow-up. Past 30-day use was asked of participants who were aware of and had ever used any e-cigarette products. Those who responded that they had ever used one or more e-cigarette products were asked, for each product they had used, “In the past 30 days, on how many of those days did you use a(n) [product]”. The list of products included: (a) a disposable electronic cigarette or vaping device that cannot be filled and recharged; (b) JUUL e-cigarettes; (c) other pod-based vaporizers such as Suorin (not including JUUL); (d) A vaping device with a tank that you refill with e-liquids but does not allow other mechanical modifications (rechargeable); (e) a vaping device or modular system that you refill with e-liquids, allows mechanical modifications, and uses your own combination of separate devices: batteries, atomizers, etc. (rechargeable); or (f) other types of e-cigarettes. Participants who reported using any of the e-cigarette products one day or more in the past month were classified as past 30-day e-cigarette users.

### 2.3. Primary Advertising Exposure and Covariate Variables

At baseline, all participants were asked if they recalled seeing or hearing any advertisements or other content related to any electronic vaping product though several different media channels in the past three months. The specific channels included regular television programming, retail stores, and online or on social media (both advertising and other online content). Participants selected each tobacco and nicotine product from a list that they had seen or heard advertising or other content for in the past three months. The items were worded as follows: “In the past 3 months, do you recall seeing any advertisements when you were watching regular television programming for any of the following?”, “In the past 3 months, do you recall seeing any advertisements at retail stores including gas stations, convenience stores, and drug stores for any of the following?”, “In the past 3 months, do you recall seeing any advertisements online or on social media (such as Facebook, YouTube, Twitter, or Instagram) when you were either reading, browsing pictures, or watching/streaming videos for any of the following?”, and “In the past 3 months, do you recall reading, seeing pictures, or watching videos related to any of the following products (not necessarily advertisements) online or on social media (such as Facebook, YouTube, Twitter, or Instagram)?”. Response options included the following e-cigarette-related products or locations: “electronic cigarettes (such as Blu, Vuse, MarkTen e-cigarettes)”, “vape pens and vape mods (box mods, tube mods, or other types of vape mods or vaporizers)”, “JUUL and other types of pod vaporizers”, “vape shops”, and “e-juice, e-liquid or e-fill (liquid used to refill an e-cigarette or vaping product)”.

Covariates included sex, race/ethnicity, household income, and U.S. region of residence and were obtained from profile surveys administered to panelists. Baseline cigarette smoking and baseline ever use of e-cigarettes were used as additional control variables. Youth under the age of 18 who had ever smoked a cigarette and who responded that they currently smoke cigarettes “every day” or “some days” at baseline and all participants 18 and older who responded that they currently smoke cigarettes “every day” or “some days” at baseline were considered current cigarette smokers.

### 2.4. Statistical Analysis

All analyses were conducted using Stata version 15.1 (StataCorp LLC, College Station, TX, USA) to obtain weighted point estimates and 95% confidence intervals for baseline sample characteristics, past 30-day e-cigarette use at follow-up, and percentage exposed to various channels of e-cigarette advertising and content. Associations among exposure to baseline advertising/product content, cigarette smoking, ever use of e-cigarettes, participant characteristics and past 30-day e-cigarette use at follow-up were measured by weighted multivariable logistic regression using svy: logit command in Stata.

The combined data set of AmeriSpeak and KnowledgePanel contained study-specific base sampling weights derived using a combination of the final panel weight and the probability of selection associated with the sampled panel member. These weights were adjusted to account for survey non-respondents to decrease potential nonresponse bias associated with sampled panel members who did not complete the survey interview for the study. The nonresponse adjusted survey weights for the study were then adjusted via a raking ratio method to external population totals (using Census data) associated with the following socio-demographic characteristics: age, sex, education, race/Hispanic ethnicity, and Census Division, to create the final study weights. Raking and re-raking were done during the weighting process such that the weighted demographic distribution of the survey completely resembled the demographic distribution in the target population. Final study weights were raked to external benchmarks on “ever use of e-cigarettes”, using the 2017 National Youth Tobacco Survey for the teen, and the 2017 National Health Interview Survey for the adult benchmarks of e-cigarette use, respectively, creating a study-specific-post-stratification weight variable. This weight variable was used in all analyses to adjust for sources of sampling and non-sampling error and allow the generalizability of the results to a national level.

## 3. Results

### 3.1. Participant Characteristics and E-Cigarette Advertisement Exposures at Baseline

Baseline sample characteristics, including sex, race/ethnicity, household income, geographic region, baseline cigarette smoking, and baseline ever use of e-cigarettes are displayed in Table 1. Among a total of 2304 youth and young adults who retained at the follow-up survey and were not past 30-day e-cigarette users at baseline, 1490 (63.1%) were youth aged 13–20, and 814 (36.9%) were young adults aged 21–24. Among youth, almost half of them were male and about one in five of youth came from a household with annual income less than USD 25,000. Among young adults, 45.7% of them were male and nearly one third had a household income less than USD 25,000.

Among retained participants at follow-up who were not past 30-day e-cigarette users at baseline, 7.9% of youth (95% CI: 6.0, 10.2) and 11.1% of young adults (95% CI: 8.4, 14.5) became past 30-day e-cigarette users. More than half of youth (63.7%, 95% CI: 59.8, 67.4) and young adults (58.3%, 95% CI: 53.6, 62.8) reported baseline exposure to any marketing content via TV, retail, or online/social media channels (Table 1).

### 3.2. Bivariate Association between Participant Characteristics and Any Advertisement Exposure

Compared with those who reported no baseline exposure to e-cigarette advertising via TV, retail, or online/social media channels, YYAs reporting any baseline e-cigarette marketing exposure via TV, retail, or online/social media were more likely to report e-cigarette use at follow-up (12.2%, 95% CI: 10.0, 15.0) than those not exposed to baseline marketing (4.1%, 95% CI: 2.5, 6.7). A greater percentage of Black, non-Hispanic respondents (17.0%, 95% CI: 14.3, 20.1) and Asian respondents (4.4%, 95% CI: 3.2, 5.9) reported baseline marketing exposure than no exposure (8.6%, 95% CI: 6.4, 11.5 and 2.0%, 95% CI: 1.3, 3.1, respectively). Those reporting ever using e-cigarettes at baseline were also more likely to report baseline marketing exposure (12.3%, 95% CI: 9.8, 15.3) than to report no exposure (5.7%, 95% CI: 3.6, 8.8.). There were no other significant differences in baseline characteristics between those exposed to baseline e-cigarette advertising and those reporting no advertising exposure (Table 2).

### 3.3. Associations between Baseline E-Cigarette Marketing Exposure and Past 30-Day E-Cigarette Use at Follow-Up: Youth

Adjusting for baseline cigarette smoking and ever use of e-cigarettes, sex, race/ethnicity, household income, and geographic region, youth reporting any baseline exposure to e-cigarette marketing had more than twice the odds of becoming a past 30-day e-cigarette user at follow-up (adjusted odds ratio (aOR): 2.77, 95% CI: 1.23, 6.24, *p* < 0.05), compared with those reporting no baseline marketing exposure (Table 3, Model 1). Youth reporting baseline exposure to e-cigarette marketing on TV (aOR: 3.08, 95% CI: 1.78, 5.33), through point-of-sale (aOR: 2.22, 95% CI: 1.18, 4.18), or online/social media (aOR: 2.33, 95% CI: 1.24, 4.39) also had increased odds of becoming a past 30-day e-cigarette user at follow-up compared to those reporting no e-cigarette marketing exposure via each of those channels (Table 3, Models 2–4). Current cigarette smoking at baseline (Model 1 aOR: 7.01, 95% CI: 1.86, 26.41; Model 2 aOR: 7.88, 95% CI: 2.11, 29.34; Model 3 aOR: 6.63, 95% CI: 1.74, 25.31; Model 4 aOR: 17.23, 95% CI: 4.64, 63.60) and ever use of e-cigarettes at baseline (Model 1 aOR: 2.87, 95% CI: 1.13, 7.23; Model 2 aOR: 3.61, 95% CI: 1.46, 8.94; Model 3 aOR: 2.68, 95% CI: 1.05, 6.81; Model 4 aOR: 3.58, 95% CI: 1.51, 8.47) also predicted past 30-day e-cigarette use at follow-up, regardless of exposure to baseline e-cigarette marketing. Residents of the Midwest had lower odds of becoming past 30-day e-cigarette users at follow-up (Model 1 aOR: 0.43, 95% CI: 0.19, 0.97; Model 2 aOR: 0.42, 95% CI: 0.18, 0.97; Model 3 aOR: 0.43, 95% CI: 0.19, 0.97; Model 4 aOR: 0.39, 95% CI: 0.16, 0.94) compared with residents of the Northeast.

### 3.4. Associations between Baseline E-Cigarette Marketing Exposure and Past 30-Day E-Cigarette Use at Follow-Up: Young Adults

Similar to results for youth, adjusting for baseline cigarette smoking, ever use of e-cigarettes, sex, race/ethnicity, household income, and geographic region, young adults reporting any baseline exposure to e-cigarette marketing had increased odds of becoming a past 30-day e-cigarette user at follow-up (aOR: 2.34, 95% CI: 1.08, 5.11, *p* < 0.05), compared with those reporting no baseline marketing exposure (Table 4, Model 1). Young adults reporting baseline exposure to e-cigarette marketing on TV (aOR: 2.48, 95% CI: 1.29, 4.80), through point-of-sale (aOR: 2.02, 95% CI: 1.00, 4.10), or online/social media (aOR: 2.18, 95% CI: 1.10, 4.32) also had increased odds of becoming a past 30-day e-cigarette user at follow-up compared to those reporting no e-cigarette marketing exposure via each of those channels (Table 4, Models 2–4). Current cigarette smokers at baseline (Model 1 aOR: 4.77, 95% CI: 1.62, 14.05; Model 2 aOR: 5.62, 95% CI: 2.08, 15.23; Model 3 aOR: 5.05, 95% CI: 1.77, 14.47; Model 4 aOR: 6.76, 95% CI: 2.35, 19.40), Black, non-Hispanic respondents (Model 1 aOR: 2.82, 95% CI: 1.18, 6.77; Model 3 aOR: 3.30, 95% CI: 1.26, 8.61; Model 4 aOR: 3.06, 95% CI: 1.12, 8.32) and Hispanic respondents (Model 1 aOR: 2.38, 95% CI: 1.05, 5.40; Model 3 aOR: 3.00, 95% CI: 1.18, 7.58) all had greater odds of becoming past 30-day e-cigarette users, while females (Model 1 aOR: 0.48, 95% CI: 0.25, 0.90; Model 2 aOR: 0.44, 95% CI: 0.23, 0.84; Model 3 aOR: 0.44, 95% CI: 0.22, 0.86; Model 4 aOR: 0.43, 95% CI: 0.21, 0.86) and young adults with household incomes of at least USD 100,000 per year (Model 1 aOR: 0.19, 95% CI: 0.05, 0.70; Model 2 aOR: 0.16, 95% CI: 0.04, 0.61; Model 3 aOR: 0.11, 95% CI: 0.02, 0.51; Model 4 aOR: 0.04, 95% CI: 0.01, 0.12) had lower odds of becoming past 30-day e-cigarette users (Table 4).

## 4. Discussion

This study contributes to growing research assessing the impact of e-cigarette advertising and marketing on e-cigarette use among YYAs. Our findings suggest that exposure to e-cigarette-related marketing and advertising was associated with an increase in subsequent e-cigarette use among YYAs in the United States. These associations persist even after controlling for baseline current use of cigarettes and ever use of e-cigarettes. We found that, among both youth and young adults, reporting baseline exposure to e-cigarette-related advertising on TV, at point-of-sale, or on social media/online was associated with an increase in past 30-day e-cigarettes use 12 months later (Youth aOR: 2.77, 95% CI: 1.23, 6.24 Young adults aOR: 2.34, 95% CI: 1.08, 5.11). Prior cross-sectional studies of the association between e-cigarette advertising exposure and past 30-day e-cigarette use also showed positive associations, but the magnitude of associations was generally smaller, with aORs generally less than 2.0 [12,17,18,40].

The majority of youth (63.7%, 95% CI: 59.8, 67.4) and young adults (58.3%, 95% CI: 53.6, 62.8) in our study reported baseline exposure to e-cigarette marketing on TV, retail, or social media/online. Exposure was highest for retail store advertising, with over half of youth (52.2%, 95% CI: 48.2, 56.3) and young adults (53.4%, 95% CI: 48.6, 58.2) reporting retail advertising exposure. Nearly as high was self-reported e-cigarette advertising or content exposure online or on social media (youth: 42.9%, 95% CI: 38.7, 47.2; young adults: 43.2%, 95% CI: 38.4, 48.1). Exposure to e-cigarette advertising on TV was also significant (youth 24.3%, 95% CI: 21.1, 27.8; young adults 22.5%, 95% CI: 19.0, 26.4). This finding is consistent with a prior study [46]. This copious exposure indicates that a large number of American YYAs were exposed to e-cigarette advertising.

Not only was overall exposure to e-cigarette advertising associated with use of e-cigarettes, but exposure via each of the individual channels we measured was also found to be associated with an increase in past 30-day e-cigarette use at follow-up, among both youth and young adults. Of the three individual channels measured, the magnitude of the association between exposure to television advertising and e-cigarette use was the largest among both youth and young adults, followed by the magnitude of association between social media/online ads exposure and e-cigarette use at follow-up, with the magnitude of association between advertising exposure at point-of-sale channels and e-cigarette use at follow-up being the least. One previous study considered amounts of advertising exposure via different channels separately; however, it did not examine the impact of each individual channel on e-cigarette use [12]. Others did analyze the impact of separate channels on use, finding some differences, but these studies were cross-sectional, used data from 2014, and found weaker associations between advertising exposure and use [17,18]. Nicksic et al. conducted a cohort study that considered the impact of each channel separately and found the strongest associations for retail store and internet advertising and current use [29]. Loukas et al. similarly found associations between several channels and e-cigarette initiation [34]. However, these data were from a period prior to the recent rapid increase in youth vaping. Our findings provide updated evidence on the association between e-cigarette advertising exposure and subsequent use of e-cigarettes by advertising channels.

Our analysis also reveals a positive association between baseline cigarette smoking or having ever used e-cigarettes and past 30-day use of e-cigarettes at 12-month follow-up. Youth who smoked cigarettes at baseline were more than six times as likely to become past 30-day e-cigarette users at follow-up, compared with those who did not report cigarette smoking at baseline; similarly, youth who had ever used e-cigarettes at baseline were roughly three times as likely to become past 30-day e-cigarette users one year later, compared with those who reported never used e-cigarettes at baseline. Although the same pattern held true for young adults who were baseline cigarette smokers, baseline ever use of e-cigarettes among young adults did not consistently predict past 30-day e-cigarette use one year later. This may be because many of those young adults who experimented with e-cigarettes before and discontinued their use were not completely satisfied by those e-cigarettes they used [47,48].

Although we did not find significant demographic differences in youth use of e-cigarettes at follow-up, young adults who were male, Black, Hispanic, or low income had higher likelihood of becoming past 30-day e-cigarette users at follow-up. These results are consistent with prior research characterizing demographics of adult e-cigarette users [3,49,50]. In the case of income, young adults living in households with income of less than USD 25,000 had significantly higher likelihood of reporting past 30-day e-cigarette use at follow-up than did young adults with incomes of at least USD 100,000. Young adults with high household incomes might reside with parents or other family members as they finish their education or begin launching their own careers. Given the association between education and income, their parents were less likely to be smokers or e-cigarette users [49], as such, this family association, or parental influence, may possibly offer a protective effect against e-cigarette use.

The high level of e-cigarette advertising exposure among American youth and young adults (YYAs) reported in our study suggests that the ubiquity of industry-sponsored e-cigarette advertising resulted in a high level of awareness of these products among YYAs, which could in turn influence their perception and attitude towards these products, and subsequently increase the likelihood of using such products among YYAs. This suggests that policies and regulations that restrict youth-oriented e-cigarette advertising may be warranted. Policies to counteract the impact of industry-sponsored e-cigarette advertising can include: (1) prohibiting e-cigarette advertising on TV (same as that for combustible cigarettes); (2) prohibiting point-of-sale e-cigarette advertising; (3) regulating and restricting e-cigarette advertising online or on social media, including prohibiting using social media influencers, prohibiting misleading claims, banning using youth and young adult models in adverting images, and adopting age-verification systems; and (4) conducting education campaigns that accurately communicate the risks of e-cigarettes to YYAs.

Our study has limitations. We only examined the e-cigarette advertising exposure through three popular channels (television, point-of-sale, and social media/online content), there may be advertising on other channels that may influence past 30-day e-cigarette use among YYAs. In addition, our study did not examine the association between the frequency of exposure to marketing and the types of advertising messages being used and e-cigarette use at follow-up. It’s possible that more frequent advertising exposure and YYA-targeted marketing messages may be more strongly associated with subsequent e-cigarette use. Furthermore, we did not control for all potential confounders, such as peer influence or social access [51], in our analyses due to lack of such measures in our survey [52]. Additionally, measures of advertising exposure and e-cigarette use were self-reported exposure in the past 3 months, which may introduce recall bias. Finally, we only used past-30-day e-cigarette use as the outcome variable which did not fully capture the intensity of e-cigarette use among YYAs. Future studies can build on our study by addressing these limitations.

## 5. Conclusions

Despite these limitations, our study provides new insight into the relationships between exposure to e-cigarette advertising and e-cigarette use among American YYAs. Our analysis suggests that e-cigarette advertising exposure via several distinct channels is associated with subsequent e-cigarette use among those who were not past 30-day e-cigarette users at baseline. Additionally, our study shows demographic differences in young adults who become past 30-day e-cigarette users at follow-up. The findings from our study suggests that restrictions on e-cigarette advertising targeted at YYAs may have the potential to reducing the likelihood of e-cigarette use among youth and young adults, which in turn, could help curb the youth vaping epidemic in the U.S.

## Figures and Tables

**Table 1 ijerph-19-12640-t001:** E-cigarette use, advertising exposures, and participant characteristics.

	Youths (13–20) *n* = 1490		Young Adults (21–24) *n* = 814	
**Baseline sample characteristics:**	*n* (wt.%)	95% CI	*n* (wt.%)	95% CI
**Sex**				
Male	702 (49.2)	45.2, 53.1	286 (45.7)	41.1, 50.3
Female	787 (50.8)	46.9, 54.8	528 (54.3)	49.7, 58.9
**Race/Ethnicity**				
White, NH	871 (55.8)	51.9, 59.6	316 (56.7)	52.3, 61.1
Black, NH	186 (13.2)	10.8, 16.1	182 (14.0)	11.4, 17.2
Other/2 +races, NH	96 (6.9)	5.0, 9.3	63 (6.4)	4.6, 8.8
Hispanic	266 (20.5)	17.5, 23.9	185 (20.0)	16.8, 23.6
Asian, NH	71 (3.6)	2.7, 4.9	68 (2.9)	1.9, 4.5
**Household Income**				
<USD 25,000	255 (19.9)	16.6, 23.6	278 (30.3)	26.3, 34.5
USD 25,000–USD 49,999	294 (19.7)	16.7, 23.2	208 (25.5)	21.7, 29.7
USD 50,000–USD 99,999	534 (31.9)	28.5, 35.5	194 (24.7)	21.0, 28.8
USD 100,000+	407 (28.5)	25.3, 32.0	134 (19.5)	16.1, 23.5
**Region**				
Northeast	247 (18.0)	14.9, 21.5	121 (14.5)	11.6, 17.9
Midwest	397 (23.6)	20.6, 26.9	143 (21.5)	18.0, 25.4
South	544 (35.6)	32.0, 39.5	376 (38.4)	34.1, 42.8
West	302 (22.8)	19.7, 26.2	174 (25.7)	21.8, 30.0
**Ever used e-cigarettes**	57 (7.6)	5.5, 10.6	134 (12.8)	10.1, 15.9
**Current cigarette smoker**	28 (3.5)	2.1, 5.8	49 (6.8)	4.5, 10.1
**E-cigarette user at follow-up**	119 (7.9)	6.0, 10.2	97 (11.1)	8.4, 14.5
**Exposure to any TV, retail or online/social media ads or content at baseline**	917 (63.7)	59.8, 67.4	547 (58.3)	53.6, 62.8
TV advertising	376 (24.3)	21.1, 27.8	229 (22.5)	19.0, 26.4
Retail store advertising	695 (52.2)	48.2, 56.3	445 (53.4)	48.6, 58.2
Online/social media advertising/content	545 (42.9)	38.7, 47.2	361 (43.2)	38.4, 48.1

wt. = weighted. CI = confidence interval.

**Table 2 ijerph-19-12640-t002:** Participant characteristics by e-cigarette advertising exposures.

	Any Exposure to TV, Retail, Online/Social Media Ad/Content			No Exposure to TV, Retail, Online/Social Media Ad/Content			
	*n*	Wt.%	95% CI	*n*	Wt.%	95% CI	*p* *
**E-cigarette use at follow-up**							<0.001
No	1273	87.8	85.0, 90.1	760	95.9	93.3, 97.5	
Yes	188	12.2	10.0, 15.0	27	4.1	2.5, 6.7	
**Baseline Characteristics:**							
**Cigarette smoker**							0.07
No	1392	94.0	91.5, 95.8	775	97.1	94.0, 98.6	
Yes	65	6.0	4.2, 8.5	11	2.9	1.4, 6.0	
**E-cigarette ever user**							0.002
No	1305	87.7	84.7, 90.2	761	94.3	91.2, 96.4	
Yes	159	12.3	9.8, 15.3	30	5.7	3.6, 8.8	
**Sex**							0.83
Male	624	47.3	43.5, 51.2	344	48.0	43.0, 53.0	
Female	839	52.7	48.9, 56.5	447	52.1	47.0, 57.0	
**Age groups**							0.08
13–20	917	65.1	61.6, 68.5	539	59.8	54.9, 64.6	
21–24	547	34.9	31.5, 38.4	252	40.2	35.4, 45.1	
**Race/Ethnicity**							<0.001
White, NH	679	50.1	46.3, 53.9	469	63.7	59.0, 68.2	
Black, NH	287	17.0	14.3, 20.1	78	8.6	6.4, 11.5	
Other/2+ races, NH	97	6.5	4.8, 8.8	61	7.5	5.2, 10.6	
Hispanic	297	22.0	19.0, 25.5	149	18.2	15.0, 22.0	
Asian, NH	104	4.4	3.2, 5.9	34	2.0	1.3, 3.1	
**Household Income**							0.52
<USD 25,000	374	25.7	22.3, 29.4	155	22.2	18.2, 26.8	
USD 25,000–USD 49,999	337	22.1	19.3, 25.2	158	21.1	16.9, 26.0	
USD 50,000–USD 99,999	433	28.6	25.3, 32.1	281	30.7	26.6, 35.1	
USD 100,000+	320	23.7	20.7, 27.0	197	26.1	22.1, 30.6	
**Region**							0.26
Northeast	242	17.0	14.3, 20.0	121	17.2	13.2, 22.1	
Midwest	309	20.6	17.8, 23.7	212	24.9	21.0, 29.2	
South	621	39.1	35.5, 42.9	287	33.6	29.2, 38.3	
West	292	23.3	20.2, 26.8	171	24.3	20.4, 28.8	

* *p*-values obtained from chi-square tests.

**Table 3 ijerph-19-12640-t003:** Effect of e-cigarette advertising exposure on past 30-day e-cigarette use among youth.

	Model 1 (1447)	Model 2 (1400)	Model 3 (1363)	Model 4 (1247)
**Any exposure to TV, retail, or online/social media ads or content**				
No (Reference group)	1.00			
Yes	**2.77 (1.23, 6.24) ***			
**Exposure to e-cigarette product Ads through TV- programming channels**				
No (Reference group)		1.00		
Yes		**3.08 (1.78, 5.33) *****		
**Exposure to e-cigarette product Ads through Point-of-sale channels**				
No (Reference group)			1.00	
Yes			**2.22 (1.18, 4.18) ***	
**Exposure to e-cigarette product Ads through online or social media channels**				
No (Reference group)				1.00
Yes				**2.33 (1.24, 4.39) ****
**Current use cigarettes at baseline**				
No (Reference group)	1.00	1.00	1.00	1.00
Yes	**7.01 (1.86, 26.41) ****	**7.88 (2.11, 29.34) ****	**6.63 (1.74, 25.31) ****	**17.23 (4.67, 63.60) *****
**Ever use of e-cigarette at baseline**				
No (Reference group)	1.00	1.00	1.00	1.00
Yes	**2.87 (1.14, 7.23) ***	**3.61 (1.46, 8.94) ****	**2.68 (1.05, 6.81) ***	**3.58 (1.51, 8.47) ****
**Sex**				
Male (Reference group)	1.00	1.00	1.00	1.00
Female	0.87 (0.47, 1.62)	0.93 (0.50, 1.72)	0.87 (0.47, 1.61)	0.83 (0.43, 1.61)
**Race/Ethnicity**				
White, NH (Reference)	1.00	1.00	1.00	1.00
Black, NH	1.74 (0.51, 5.92)	1.47 (0.46, 4.75)	1.76 (0.53, 5.83)	1.59 (0.45, 5.69)
Other/2+ races, NH	1.40 (0.50, 3.89)	1.55 (0.57, 4.21)	1.38 (0.50, 3.85)	1.63 (0.60, 4.43)
Hispanic	1.55 (0.74, 3.24)	1.61 (0.79, 3.31)	1.52 (0.73, 3.18)	1.35 (0.62, 2.93)
Asian, NH	0.69 (0.20, 2.35)	0.57 (0.17, 1.99)	0.86 (0.26, 2.84)	0.34 (0.10, 1.17)
**Household Income**				
<USD 25,000 (Reference)	1.00	1.00	1.00	1.00
USD 25,000–USD 49,999	2.41 (0.99, 5.86)	**2.77 (1.13, 6.79) ***	2.33 (0.96, 5.65)	2.50 (0.95, 6.55)
USD 50,000–USD 99,999	1.73 (0.64, 4.68)	2.15 (0.84, 5.54)	1.77 (0.66, 4.74)	2.36 (0.85, 6.61)
USD 100,000+	2.38 (0.88, 6.47)	2.64 (0.99, 7.07)	2.38 (0.89, 6.40)	2.22 (0.72, 6.86)
**Region**				
Northeast (Reference)	1.00	1.00	1.00	1.00
Midwest	**0.43 (0.19, 0.97) ***	**0.42 (0.18, 0.97) ***	**0.43 (0.19, 0.97) ***	**0.39 (0.16, 0.94) ***
South	0.57 (0.22, 1.46)	0.54 (0.21, 1.42)	0.58 (0.23, 1.50)	0.59 (0.22, 1.55)
West	0.56 (0.24, 1.33)	0.55 (0.23, 1.32)	0.57 (0.24, 1.36)	0.52 (0.20, 1.35)

* *p* ≤ 0.05, ** *p* ≤ 0.01, *** *p* ≤ 0.001.

**Table 4 ijerph-19-12640-t004:** Effect of e-cigarette advertising exposure on past 30-day e-cigarette use among young adults.

	Model 1 (791)	Model 2 (776)	Model 3 (735)	Model 4 (690)
**Any exposure to TV, retail, or online/social media ads or content**				
No (Reference group)	1.00			
Yes	**2.34 (1.08, 5.11) ***			
**Exposure to e-cigarette product Ads through TV-programming channels**				
No (Reference group)		1.00		
Yes		**2.48 (1.29, 4.80) ****		
**Exposure to e-cigarette product Ads through Point-of-sale channels**				
No (Reference group)			1.00	
Yes			**2.02 (1.00, 4.10) ***	
**Exposure to e-cigarette product Ads through online or social media channels**				
No (Reference group)				1.00
Yes				**2.18 (1.10, 4.32) ***
**Current use cigarettes at baseline**				
No (Reference group)	1.00	1.00	1.00	1.00
Yes	**4.77 (1.62, 14.05) ****	**5.62 (2.08, 15.23) ****	**5.05 (1.77, 14.47) ****	**6.76 (2.35, 19.40) *****
**Ever use of e-cigarette at baseline**				
No (Reference group)	1.00	1.00	1.00	1.00
Yes	2.15 (0.98, 4.70)	**2.70 (1.22, 5.97) ***	2.21 (0.98, 4.97)	1.64 (0.70, 3.82)
**Sex**				
Male (Reference group)	1.00	1.00	1.00	1.00
Female	**0.48 (0.25, 0.90) ***	**0.44 (0.23, 0.84) ***	**0.44 (0.22, 0.86) ***	**0.43 (0.21, 0.86) ***
**Race/Ethnicity**				
White, NH (Reference)	1.00	1.00	1.00	1.00
Black, NH	**2.82 (1.18, 6.77) ***	2.41 (0.99, 5.84)	**3.30 (1.26, 8.61) ***	**3.06 (1.12, 8.32) ***
Other/2+ races, NH	0.84 (0.27, 2.67)	0.70 (0.21, 2.26)	0.91 (0.27, 3.08)	0.91 (0.24, 3.42)
Hispanic	**2.38 (1.05, 5.40) ***	2.39 (1.00, 5.72)	**3.00 (1.18, 7.58) ***	2.57 (0.97, 6.81)
Asian, NH	1.85 (0.65, 5.32)	1.52 (0.51, 4.48)	2.38 (0.78, 7.27)	2.35 (0.76, 7.27)
**Household Income**				
<USD 25,000 (Reference)	1.00	1.00	1.00	1.00
USD 25,000–USD 49,999	0.45 (0.20, 1.03)	0.44 (0.19, 1.05)	0.43 (0.18, 1.03)	0.42 (0.17, 1.01)
USD 50,000–USD 99,999	1.08 (0.51, 2.28)	1.03 (0.48, 2.20)	0.89 (0.41, 1.92)	0.88 (0.40, 1.93)
USD 100,000+	**0.19 (0.05, 0.70) ***	**0.16 (0.04, 0.61) ****	**0.11 (0.02, 0.51) ****	**0.04 (0.01, 0.12) *****
**Region**				
Northeast (Reference)	1.00	1.00	1.00	1.00
Midwest	1.00 (0.35, 2.87)	0.66 (0.25, 1.76)	0.95 (0.31, 2.94)	0.59 (0.18, 1.89)
South	0.91 (0.38, 2.19)	0.61 (0.27, 1.38)	0.93 (0.37, 2.35)	0.86 (0.35, 2.12)
West	0.99 (0.37, 2.62)	0.66 (0.26, 1.70)	0.93 (0.33, 2.64)	0.83 (0.29, 2.40)

* *p* ≤ 0.05, ** *p* ≤ 0.01, *** *p* ≤ 0.001.

## Data Availability

Data will be available in a publicly accessible repository once the paper is accepted for publication.

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
