# Peer review of "Association between E-Cigarette Advertising Exposure and Use of E-Cigarettes among a Cohort of U.S. Youth and Young Adults"

_ijerph, 2022, doi:10.3390/ijerph191912640_

Round 1

Reviewer 1 Report

Authors address a topic that deals with a social problem - the 

e-cigarettes consumption amongst young people.

In general , the paper has a proper form but authors are advised to introduce a section that deals more with the literature review regarding other studies that have something consistent to say about e-cigarettes consumption behavior and influence of marketing stimuli over this type of consumption.

Within this section, authors should clearly define the main hypotheses of their research.

Also, in order to make the paper more accessible to readers, authors are advised to explain the meaning of every annotation and abbreviation of indices used during their analysis. (for example ....”aOR”...etc).

Author Response

Thank you very much for your instructive comments. Please see the attachment for our responses.

Reviewer 2 Report

General

The growth of e-cigarette use, particularly among young people is a significant public health issue. Advertising could have a significant role in the promotion of e-cigarette use among young people, and it is important to have up to date information on the impact of advertising to help inform policy. As such the study should help inform policy development, both in US and other countries. Overall the paper reads well. I have suggested a number of minor changes to help increase the value of the paper. In particular, I feel it would be useful to discuss the policy implications in more detail. have made the following comments to particular sections of the text and I hope the authors find this useful in making a revision.

1. Introduction

The introduction provides a good overview of the topic and rationale for the study.

Line 66

The terms youth and young adult need to be clarified by giving reference to age ranges).

2. Materials and Methods

Line 94-107

More information should be provided on the panels. Are these national panels? Are they statistically representative by age, gender, and socioeconomic group? How were the panels originally sampled? What was the process by which permission was granted to contact these people to ask if they would take part in the study? Why were two panels used? Could one person have been on both panels? Why did you contact people by web only for the knowledge panel? This may have biased sample selection. Further detail required to help clarify these issues.

2.3 Line 124

Last 3 months – this is a long timeframe to base recall of advertising. It would be useful to insert supporting literature if this is comparable to other studies of advertising. In addition this issue warrants consideration in the study limitations.

2.4 Line 153

Need to expand in terms of the variables used to weight data.

2.4 Line 158

Weighted multivariable logistic regression needs to be expanded to provide further explanation.

3. Results

3.1 Line 163-168

A title “Sociodemographic profile” may be more appropriate.  This section should just contain the sample profile of the study. Information about e-cigarette use and exposure to advertising should be moved t a separate section. This wil make the results more user friendly and easier to follow.A short description of baseline characteristics should be given with information in terms of whether it is representative at national or local level and to reaffirm the application of the weighting.

Table 3 line 211

This table needs to be made user friendly. In addition, consideration should be given to undertaking an analysis of the number of advertising channels exposed to.

4. Discussion

The discussion should be expanded to tease out the implications of the study findings. For example it would be interesting to provide a comparison of the level of exposure to e-cigarette advertising with exposure to other forms of advertising. How for example would it compare to advertising for sweets/candy products? In addition what are the implications of the large level of the large level of advertising exposure? What options should policy makers consider to counteract the impact of advertising? Should initiatives be targeted any particular sociodemographic groups?

Author Response

Thank you very much for your instructive comments. Please see the attachment for our detailed responses.

Author Response

(The authors gave the same response as above.)
